# Murine cytomegaloviruses m139 targets DDX3 to curtail interferon production and promote viral replication

Olha Puhach[1], Eleonore Ostermann[1], Christoph Krisp[2], Giada Frascaroli[1], Hartmut Schlüter[2], Melanie M. Brinkmann[3,4], Wolfram Brune[1]*

**1** Heinrich Pette Institute, Leibniz Institute for Experimental Virology, Hamburg, Germany, **2** Institute for Clinical Chemistry and Laboratory Medicine, Mass Spectrometric Proteomics, University Medical Center Hamburg-Eppendorf, Hamburg, Germany, **3** Viral Immune Modulation Research Group, Helmholtz Centre for Infection Research, Braunschweig, Germany, **4** Institute of Genetics, Technische Universität Braunschweig, Braunschweig, Germany

* wolfram.brune@leibniz-hpi.de

**Data Availability Statement:** All relevant data are within the manuscript and its Supporting Information files.

## Abstract

Cytomegaloviruses (CMV) infect many different cell types and tissues in their respective hosts. Monocytes and macrophages play an important role in CMV dissemination from the site of infection to target organs. Moreover, macrophages are specialized in pathogen sensing and respond to infection by secreting cytokines and interferons. In murine cytomegalovirus (MCMV), a model for human cytomegalovirus, several genes required for efficient replication in macrophages have been identified, but their specific functions remain poorly understood. Here we show that MCMV m139, a gene of the conserved US22 gene family, encodes a protein that interacts with the DEAD box helicase DDX3, a protein involved in pathogen sensing and interferon (IFN) induction, and the E3 ubiquitin ligase UBR5. DDX3 and UBR5 also participate in the transcription, processing, and translation of a subset of cellular mRNAs. We show that m139 inhibits DDX3-mediated IFN-α and IFN-β induction and is necessary for efficient viral replication in bone-marrow derived macrophages. In vivo, m139 is crucial for viral dissemination to local lymph nodes and to the salivary glands. An m139-deficient MCMV also replicated to lower titers in SVEC4-10 endothelial cells. This replication defect was not accompanied by increased IFN-β transcription, but was rescued by knockout of either DDX3 or UBR5. Moreover, m139 co-localized with DDX3 and UBR5 in viral replication compartments in the cell nucleus. These results suggest that m139 inhibits DDX3-mediated IFN production in macrophages and antagonizes DDX3 and UBR5-dependent functions related to RNA metabolism in endothelial cells.

## Author summary

Human cytomegalovirus is an opportunistic pathogen that causes severe infections in immunocompromised individuals. The virus infects certain cell types, such as macrophages and endothelial cells, to ensure its dissemination within the body. Little is known

**Funding:** This study was funded by the Deutsche Forschungsgemeinschaft (DFG) grant BR 1730/6-1 (WB). The Heinrich Pette Institute is supported by the Free and Hanseatic City of Hamburg and the Federal Ministry of Health. MMB was funded by the SMART BIOTECS alliance, which is supported by the Ministry of Science and Culture of Lower Saxony, Germany, and the Helmholtz Association, grant reference number W2/W3-090. The funders had no role in study design, data collection and analysis, decision to publish, or preparation of the manuscript.

**Competing interests:** The authors have declared that no competing interests exist.

about the viral factors that promote a productive infection of these cell types. The identification of critical viral factors and the molecular pathways they target can lead to the development of novel antiviral treatment strategies. Using the mouse cytomegalovirus as a model, we studied the viral m139 gene, which is important for virus replication in macrophages and endothelial cells and for dissemination in the mouse. This gene encodes a protein that interacts with the host proteins DDX3 and UBR5. Both proteins are involved in gene expression, and the RNA helicase DDX3 also participates in mounting an innate antiviral response. By interacting with DDX3 and UBR5, m139 ensures efficient viral replication in endothelial cells. Importantly, we identify m139 as a new viral DDX3 inhibitor, which curtails the production of interferon by macrophages.

## Introduction

Cytomegaloviruses (CMVs) have a broad tissue tropism and infect a wide variety of cells types. Cells of the myeloid lineage play an important role in CMV dissemination and pathogenesis: latently infected monocytes disseminate the virus to target organs [1–3]. Differentiation of monocytes into macrophages as they extravasate into tissues triggers the lytic replication cycle and the production of infectious virus, which transmits the infection to the surrounding parenchymal cells. Efficient replication in macrophages requires that the virus is able to curtail its recognition by the innate immune system [4]. This is particularly important in macrophages, as these cells express numerous sensors that activate host antiviral defenses and induce the production of cytokines and interferons [5].

Another cell type involved in CMV dissemination and pathogenesis are endothelial cells. They support productive virus replication and are thought to contribute to dissemination via the blood stream [6]. Moreover, endothelial cells have been proposed to be sites of latency [7].

Murine cytomegalovirus (MCMV) serves as a small animal model for human cytomegalovirus (HCMV). It encodes a number of genes known to be important for efficient replication in macrophages, several of which belong to the US22 gene family. This gene family comprises 12 members in both, HCMV and MCMV, that contain up to four conserved sequence motifs [8,9]. The functions of most US22 family genes remain unknown; only in a few cases has the mechanism of action been resolved. MCMV genes m142 and m143 are the only US22 family genes essential for viral replication [9]. The m142 and m143 proteins form a heterodimeric dsRNA-binding complex and inhibit the activation of the dsRNA-dependent protein kinase, PKR [10,11]. Their homologs in HCMV, TRS1 and IRS1, fulfill essentially the same function [12,13]. Inhibition of PKR is essential for CMV replication in cell culture and in vivo [14–17]. Another well-characterized US22 family gene is MCMV M36 and its homolog in HCMV, UL36. The M36 and UL36 proteins prevent apoptosis by inhibiting caspase-8 activation [9,18]. This function is particularly important for virus replication in macrophages and dissemination in vivo [19–21].

Besides M36, the MCMV genes m139, m140, and m141 promote efficient replication of the virus in macrophages [9,22–24]. The three genes are clustered within the MCMV genome and their protein products can bind to each other, suggesting that they might function cooperatively [25]. On the other hand, the three proteins appear to have individual functions. The m140 protein is required for efficient capsid assembly in macrophages [26]. However, the underlying mechanism remains incompletely understood. The m140 protein interacts with m141 and stabilizes it by protecting it from proteasomal degradation [27]. The m141 protein

directs m140 to a perinuclear region of infected macrophages adjacent to the microtubule organizing center, suggesting an involvement of aggresomes [26].

The least is known about m139. Its requirement for efficient replication in IC-21 macrophages has been documented [9], but m139 was not required for the functions described for m140 and m141 [26,27]. However, a recently performed screen for MCMV inhibitors of IFN-β induction identified m139 as a candidate [28]. These findings prompted us to investigate the function of m139, its role for viral replication in macrophages and other cell types, and its requirement for viral dissemination in vivo.

In this study, we show that m139 is required for efficient virus replication in bone marrow-derived macrophages and endothelial cells in vitro and for viral dissemination in vivo. By affinity purification and mass spectrometry, we identified the host proteins DDX3 and UBR5 as interactors of m139. Knockouts of DDX3 or UBR5 in endothelial cells and bone marrow-derived macrophages rescued the replication defect of an m139-deficient MCMV. Moreover, m139 inhibited DDX3-dependent IFN induction in macrophages to facilitate efficient virus replication in these cells. This study reveals that m139 counteracts the host factors DDX3 and UBR5 to subvert host antiviral responses and promote viral replication.

## Results

### The MCMV protein m139 is expressed with early kinetics and recruited to viral replication compartments

The m139 ORF is transcribed with early kinetics and encodes two proteins of 72 and 61 kDa [22]. To further characterize the m139 proteins in the absence of a specific antibody, we generated a recombinant MCMV expressing a C-terminally HA-tagged m139. By BAC recombineering an HA tag sequence was inserted at the 3' end of the m139 ORF. We verified the classification of m139 as an early protein by using a cycloheximide (CHX) release assay. Fibroblasts were infected in the presence of CHX, a translation inhibitor. Four hours post infection (hpi), cells were washed and incubated either with normal medium, in order to relieve the inhibition, or with medium containing actinomycin D (ActD), a transcription inhibitor, to selectively allow the translation of viral immediate-early transcripts. As shown in Fig 1A, m139 was expressed in untreated cells and after release from the CHX block, but not in the presence of ActD. Thus, m139 is expressed with early kinetics, similar to the M112-113 encoded Early 1 (E1) proteins.

To clarify the subcellular localization of m139 during infection, we used a cell fractionation assay to separate the nuclear from the cytoplasmic fraction. In fibroblasts infected with MCMV m139-HA, the m139 protein was found in both nuclear and cytoplasmic fractions at early and late times post infection (Fig 1B), consistent with a previous study [22]. By immunofluorescence, m139 was also detected in the cytoplasm and in the nucleus. It was found dispersed throughout the cytoplasm at early times and was enriched in perinuclear dots at late times post infection (Fig 1C). In the nucleus, m139 co-localized with the viral E1 proteins (Fig 1C). The MCMV E1 proteins are encoded by the M112-113 gene, exist in four isoforms [29,30], and are markers for viral replication compartments [31].

An in silico analysis of the m139 amino acid sequence revealed the presence of a nuclear export signal (NES). The predicted NES sequence SEIRVLRGVDLSD was mapped to residues 215–227. Consistent with this prediction, the m139 protein was detected exclusively in the cytoplasm of cells upon ectopic expression of m139 (S1A Fig). However, co-transfection with an E1 expression vector was sufficient to cause m139 accumulation in nuclear E1-positive dots (S1B Fig), suggesting that the viral E1 proteins recruit m139 to pre-replication compartments.

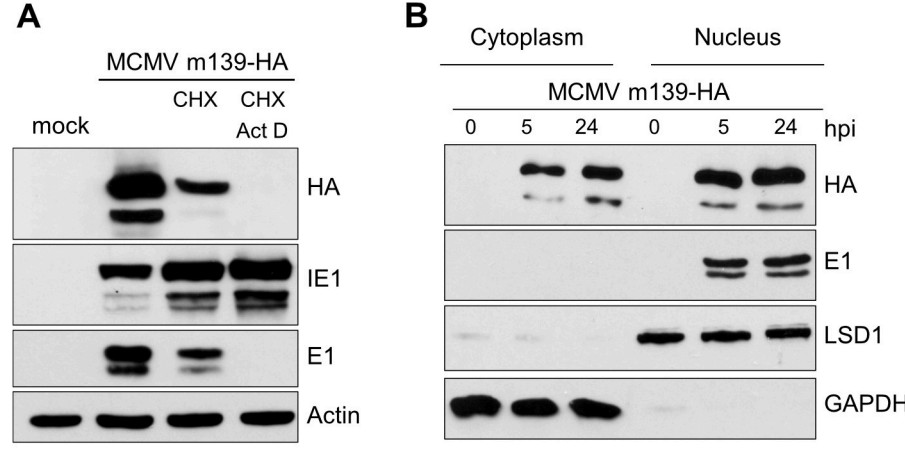

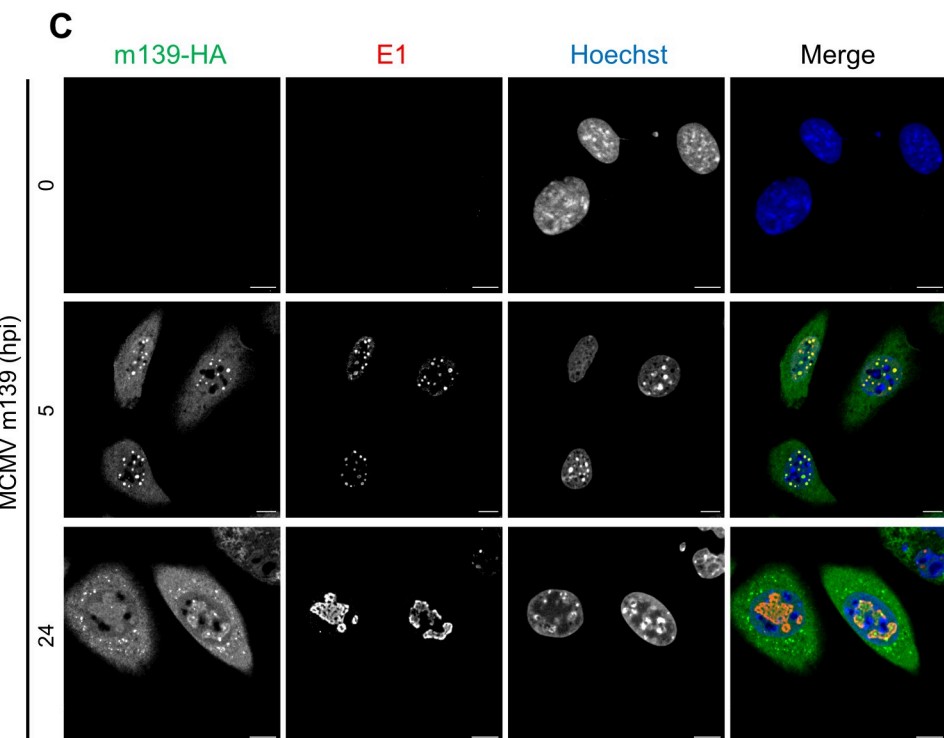

**Fig 1. MCMV m139 is expressed with early kinetics and localizes to the cytoplasm and viral replication compartments in the nucleus.** (A) 10.1 fibroblasts were infected with MCMV m139-HA (MOI = 3) in the presence of CHX (5 μg/ml) or left untreated. Four hpi, the CHX-containing medium was replaced by normal medium or medium containing ActD (50 μg/ml). After 4 hours, viral protein expression was analyzed by immunoblotting. (B) 10.1 fibroblasts were infected with MCMV m139-HA (MOI = 5). Cell lysates were harvested at 5 and 24 hpi, separated into nuclear and cytoplasmic fractions, and analyzed by immunoblotting. (C) 10.1 fibroblast were infected with MCMV m139-HA (MOI = 1), fixed at 5 and 24 hpi, and subjected to immunofluorescence. Mock-infected cells were used as a control. m139 was detected with an anti-HA antibody and the viral E1 (encoded by M112-113) proteins with an E1-specific antibody. Nuclei were counterstained with Hoechst 33342. Representative images taken by confocal microscopy are shown. Scale bar, 10 μm. The results shown in this figure are representative of two (A) or three (B, C) independent experiments.

## m139 is required for efficient MCMV replication in macrophages and endothelial cells

A previous study has reported that m139 is required for MCMV replication to high titers in IC-21 peritoneal macrophages [9]. In order to investigate the role of m139 for MCMV replication in different cell types of mouse origin, we constructed an MCMV m139 knockout mutant by introducing a stop at codon position 32 of the m139 ORF. The genome integrity of the MCMV m139-HA and m139*stop* mutants was verified by deep sequencing to rule out accidental mutations elsewhere in the genome. We compared the replication properties of the MCMV m139*stop* mutant to the wildtype MCMV m139-HA by multistep replication kinetics in 10.1 fibroblasts, TCMK-1 epithelial cells, SVEC4-10 endothelial cells, and immortalized bone marrow-derived macrophages (iBMDM). In 10.1 fibroblasts and TCMK-1 epithelial cells, the MCMV m139*stop* mutant replicated to titers comparable to MCMV m139-HA (Fig 2A and 2B). By contrast, MCMV m139*stop* replicated to substantially lower levels in SVEC4-10 endothelial cells (Fig 2C) and iBMDM (Fig 2D). These results demonstrated that m139 is required for efficient MCMV replication in specific cell types such as macrophages and endothelial cells, but is dispensable in fibroblasts and epithelial cells. MCMV m139-HA and WT MCMV replicated to similar titers in these cell types, indicating that the C-terminal HA tag did not negatively affect MCMV replication (Fig 2C and 2D).

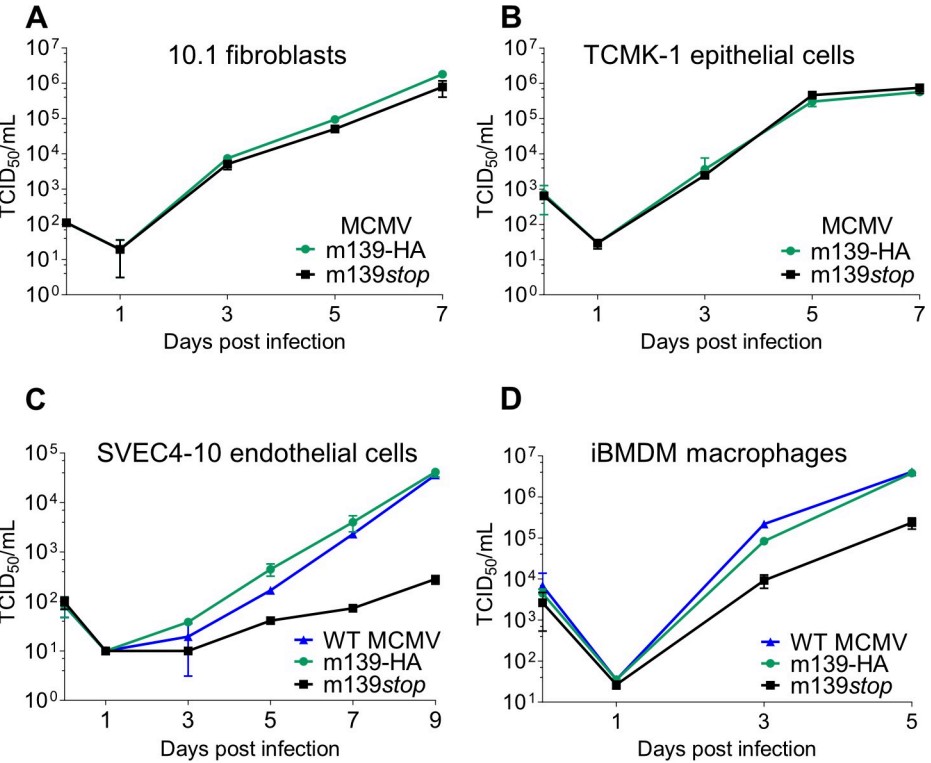

**Fig 2. MCMV m139 is important for viral replication in macrophages and endothelial cells.** Multistep replication kinetics of WT MCMV, MCMV m139-HA and MCMV m139*stop* in murine 10.1 fibroblasts (A), TCMK-1 epithelial cells (B), SVEC4-10 endothelial cells (C), and immortalized bone marrow-derived macrophages (iBMDM) (D). iBMDM were infected at an MOI of 0.025, all others at an MOI of 0.01. Virus released by infected cells into the supernatant was quantified by titration. Viral titers are shown as means ±SD of three biological replicates.

## m139 interacts with host proteins DDX3 and UBR5

In order to gain further insight into the role of m139 in MCMV replication, we wanted to identify m139-interacting proteins in MCMV-infected cells. We used stable isotope labeling of amino acids in cell culture (SILAC [32]) combined with affinity purification and mass spectrometry (AP-MS) to identify viral and host proteins interacting with m139. SVEC4-10 cells were infected with MCMV m139-HA or WT MCMV, and m139 was immunoprecipitated using an anti-HA affinity matrix. Proteins identified with less than 4-fold enrichment in the MCMV m139-HA sample or less than 2 unique peptides detected were excluded from further analysis. Using these criteria, 11 putative interaction partners of m139 were identified (Table 1). The candidate list included MCMV proteins m140 and m141, which are known interaction partners of m139 [25]. Among the host proteins, we focused on those previously reported to affect CMV replication or type I IFN signaling: the ATP-dependent RNA helicase DDX3 (a.k.a. DEAD box protein 3), the E3 ubiquitin-protein ligase UBR5 (a.k.a. EDD1), and the interferon-induced protein with tetratricopeptide repeats 1 (IFIT1) [33–37]. To verify the interactions of m139 with DDX3, UBR5, and IFIT1, m139-HA or M45-HA as control were immunoprecipitated from lysates of SVEC4-10 cells infected with MCMV m139-HA or MCMV M45-HA. As shown in Fig 3A, DDX3 and UBR5 co-precipitated with m139, but IFIT1 did not. A similar result was obtained with MCMV-infected iBMDM (S2 Fig). None of the three cellular proteins co-precipitated with M45. Interestingly, the viral E1 proteins did not co-precipitate with m139 (Fig 3A), even though the E1 proteins mediate the recruitment of m139 to nuclear replication compartments (S1 Fig). These finding suggested that the viral m139 and E1 proteins interact indirectly or through a weak interaction.

DDX3 and UBR5 are multifunctional host proteins known to participate in fundamental cellular processes such as transcription, translation, and cell proliferation [38,39]. The RNA helicase DDX3 exists in two isoforms, DDX3X and DDX3Y, which are thought to have redundant functions. While DDX3Y expression is largely confined to the male germ line, DDX3X (hereafter referred to as DDX3) is ubiquitously expressed [40]. DDX3 has multiple functions in the context of RNA metabolism due to its ability to reorganize RNA secondary structures and ribonucleoprotein (RNP) complexes. Additionally, DDX3 is also involved in pathogen sensing and interferon (IFN) activation [40,41]. UBR5 is a member of the HECT E3 ubiquitin ligase family, whose functions remain incompletely understood. Only a subset of its known

**Table 1. Proteins interacting with m139 identified by AP-MS.**

| Protein name | 1[st] replicate | | 2[nd] replicate | |
| --- | --- | --- | --- | --- |
| | Unique peptides | Enrichment(L/H) | Unique peptides | Enrichment (H/L) |
| Myosin phosphatase Rho-interacting protein (MPRIP) | 14 | 56.3 | 11 | 53.1 |
| MCMV protein m141 | 31 | 46.7 | 27 | 58.7 |
| MCMV protein m140 | 26 | 39.5 | 19 | 35.9 |
| E3 ubiquitin-protein ligase UBR5 | 12 | 14.8 | 8 | 31.3 |
| Ankycorbin (RAI14) | 20 | 18.9 | 12 | 22.5 |
| GEM-interacting protein (GMIP) | 8 | 16.4 | 5 | 20.3 |
| Interferon-induced protein with tetratricopeptide repeats 1 (IFIT1) | 13 | 7.1 | 12 | 14.7 |
| ATP-dependent RNA helicase DDX3 | 2 | 7.9 | 2 | 5.9 |
| Codanin-1 (CDAN1) | 3 | 3.7 | 3 | 7.8 |
| MCMV protein m142 | 9 | 3.1 | 12 | 2.6 |
| Unconventional myosin-Ic (MYO1C) | 2 | 2.0 | 2 | 2.3 |

Proteins labelled by SILAC with light (L) or heavy (H) amino acids and label switch between replicates.

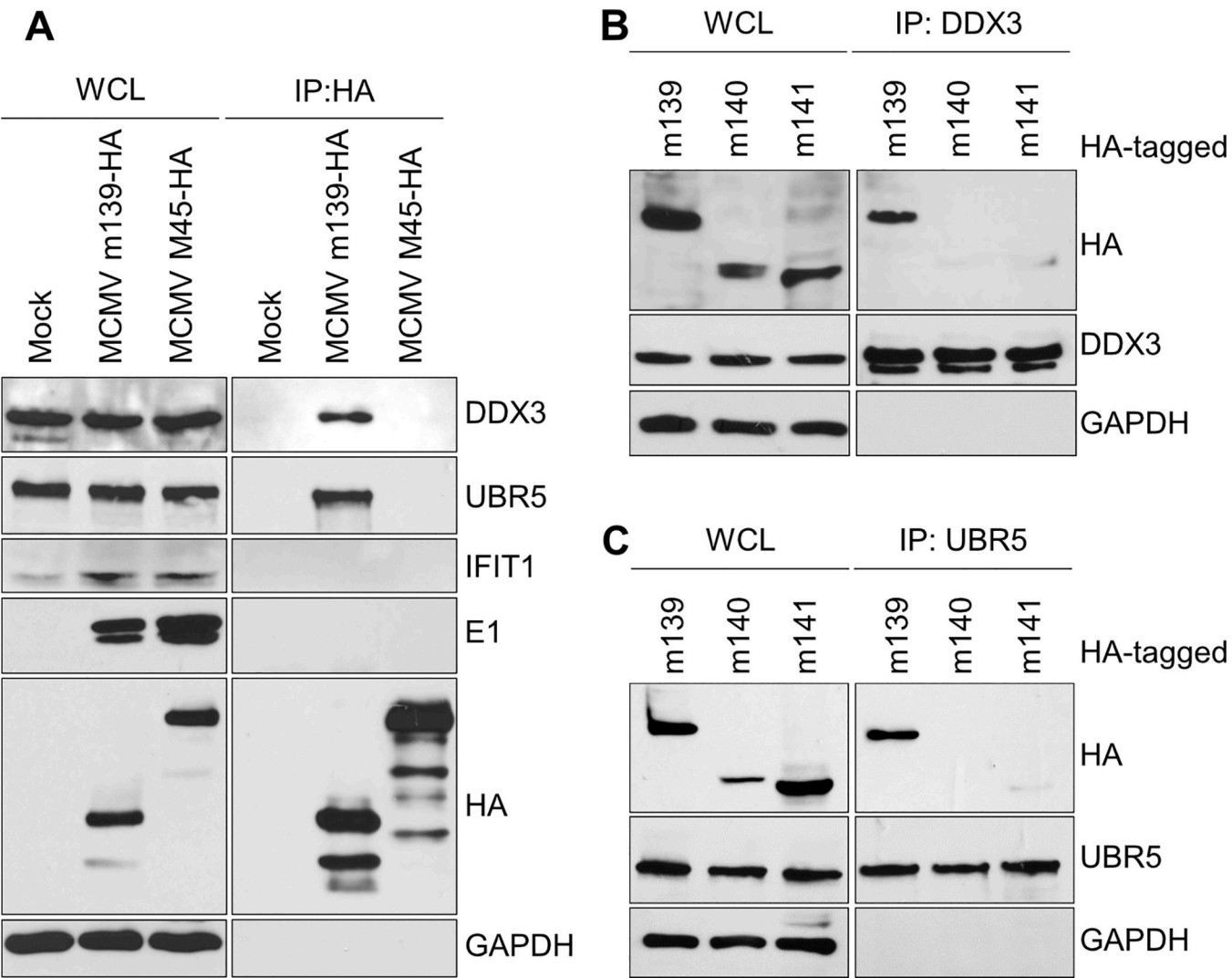

**Fig 3. m139 interacts with DDX3 and UBR5.** (A) SVEC4-10 cells were infected with MCMV m139-HA or M45-HA (MOI = 5). Cell lysates were collected 24 hpi and subjected to immunoprecipitation (IP) using an anti-HA affinity matrix. Co-precipitating proteins were detected by immunoblotting with specific antibodies. (B, C) HEK-293A cells were transfected with expression plasmids encoding HA-tagged MCMV proteins m139, m140, or m141. Cell lysates were subjected to immunoprecipitation (IP) using anti-DDX3 (B) or anti-UBR5 (C) antibodies. Co-precipitating proteins were detected by immunoblotting as in A. The results shown in this figure are representative of two (B, C) or three (A) independent experiments.

interaction partners are targeted for ubiquitination [39]. Besides its role in transcription and translation, UBR5 also regulates the DNA damage response and promotes cell cycle progression [39].

Considering that m139 can form a complex with m140 and m141 in MCMV infected cells [25], we tested whether m139 can interact with DDX3 or UBR5 in the absence of m140 and m141. For this, DDX3 or UBR5 were immunoprecipitated from lysates of HEK-239A cells transfected with HA-tagged versions of m139, m140, or m141. Under these conditions, only m139 co-precipitated with DDX3 and UBR5 (Fig 3B and 3C). This result suggests that m139 itself specifically interacts with DDX3 and UBR5 and that m140 and m141 are not required for these interactions.

Next, we tested whether m139 affects the subcellular distribution of DDX3 and UBR5 in infected cells. To do this, we infected SVEC4-10 endothelial cells with MCMV m139-HA or

MCMV m139*stop*. DDX3 showed a predominantly cytoplasmic distribution in uninfected cells. In MCMV-infected cells it also accumulated in nuclear replication compartments, and this redistribution was dependent on m139 expression (Fig 4A). UBR5 was detected only in the nucleus. It also accumulated in nuclear replication compartments of infected cells, but this redistribution was not dependent on m139 expression (Fig 4B), suggesting that the accumulation of UBR5 in viral replication compartments occurs by other means.

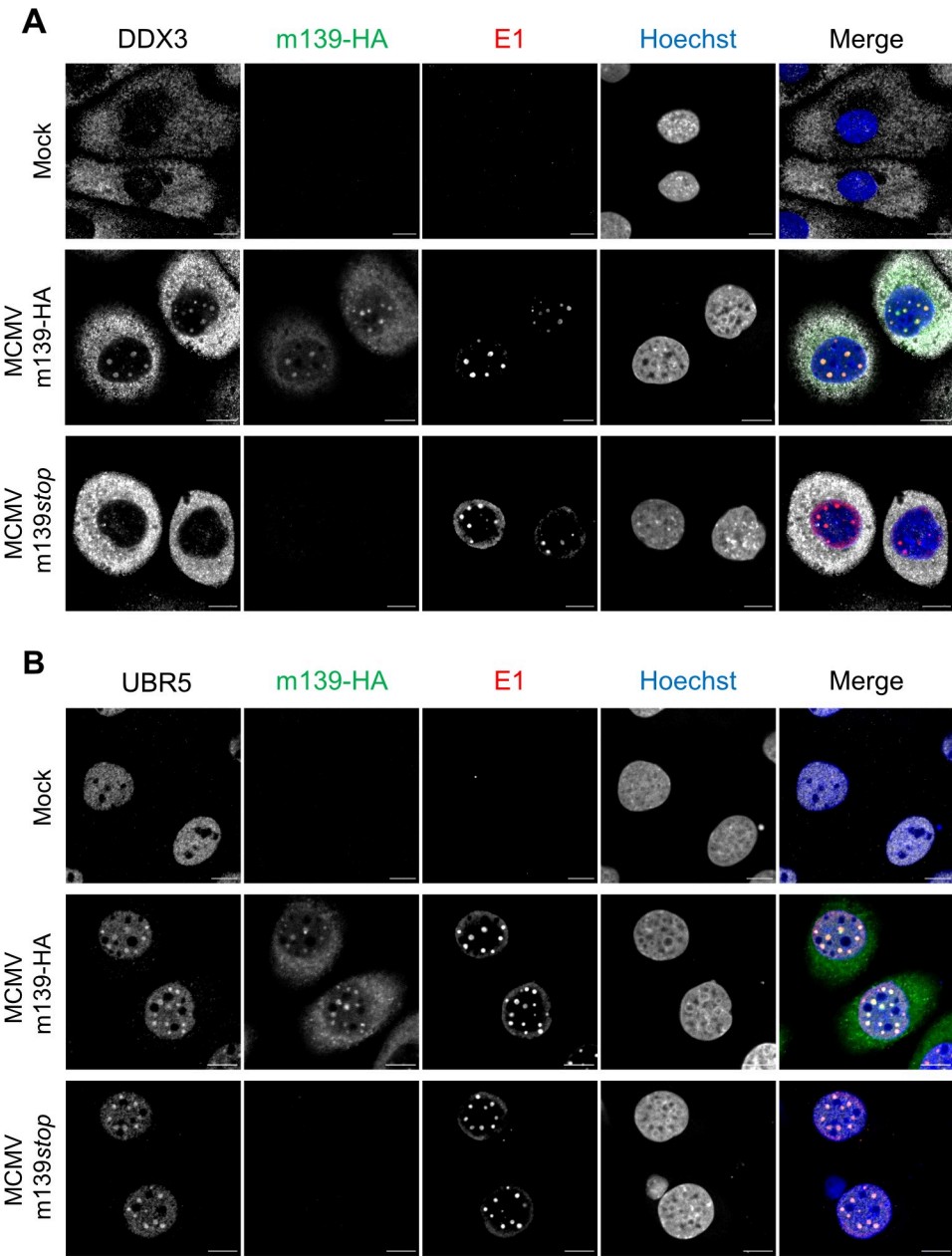

**Fig 4. DDX3 and UBR5 are recruited to viral replication compartments.** SVEC4-10 endothelial cells were infected with MCMV m139-HA or MCMV m139*stop* (MOI = 1) and fixed at 8 hpi. The subcellular localization of DDX3 (A), UBR5 (B), and the viral m139 and E1 (M112-113) proteins was analyzed by immunofluorescence using protein-specific and anti-HA antibodies. Nuclei were counterstained with Hoechst 33342. Representative images acquired by confocal microscopy are shown. Scale bar, 10 μm. The results shown in this figure are representative of three independent experiments.

## DDX3 and UBR5 restrict MCMV replication in SVEC4-10 endothelial cells

The DEAD box RNA helicase DDX3 acts as a proviral factor for several RNA viruses, but also participates in the innate immune response, thereby functioning as an antiviral factor [40,41]. As MCMV replication in SVEC4-10 endothelial cells and iBMDM was impaired in the absence of m139 (Fig 2C and 2D), we wanted to determine whether DDX3 or UBR5 are responsible for the impaired replication of the m139*stop* mutant. Therefore, we generated DDX3 and UBR5-deficient SVEC4-10 cells by CRISPR/Cas9 gene editing. Despite great efforts we were unable to obtain completely DDX3-deficient cells. However, we were able to isolate a SVEC4-10 cell clone that expresses DDX3 in very small amounts (Fig 5A). In contrast, it was not difficult to isolate UBR5-deficient SVEC4-10 cells (Fig 5B). Remarkably, the replication defect of MCMV m139*stop* in WT SVEC4-10 cells (Fig 5C) could be rescued to WT levels in DDX3-deficient (Fig 5D) as well as in UBR5-deficient cells (Fig 5E). These findings suggested that DDX3 and UBR5 act as restriction factors of MCMV replication in SVEC4-10 cells and that m139 counteracts this restriction.

As both, DDX3 and UBR5, were involved in restraining the replication of MCMV m139*stop*, we interrogated whether the interaction of m139 with these two proteins is interdependent. We tested this in immunoprecipitation experiments with DDX3 and UBR5-deficient cells. Indeed, reduced amounts of UBR5 co-precipitated with m139 in DDX3-deficient cells (Fig 5F), and reduced amounts of DDX3 co-precipitated with m139 in UBR5-deficient cells (Fig 5G). These results indicate that the interaction of m139 with UBR5 is dependent on DDX3 and vice versa. It is conceivable that m139 forms a heterotrimeric complex with DDX3 and UBR5 within viral replication compartments.

Considering that UBR5 is an E3 ubiquitin ligase that can mark interacting proteins for proteasomal degradation, we tested whether m139 induces DDX3 degradation by recruiting UBR5. We infected SVEC40-10 and iBMDM with MCMV m139-HA and m139*stop* and analyzed DDX3 levels over time. As shown in S3 Fig, m139 expression by MCMV did not induce decreased DDX3 levels. Thus, the recruitment of UBR5 by m139 does not seem to induce DDX3 degradation.

## m139 inhibits DDX3-mediated IFN induction in macrophages and in MCMV-infected mice

DDX3 is known to function as a mediator of the IFN-β antiviral response [40,41]. Moreover, a recent transfection-based screen for MCMV-encoded inhibitors of IFN-β induction identified m139 as one of several candidates [28]. To examine the effect of m139 on IFN-β induction we used immortalized bone marrow-derived macrophages expressing firefly luciferase under control of the endogenous IFN-β promoter (iBMDM β-luc). Infection of these cells with MCMV m139-HA resulted in an increase in luciferase expression, confirming that macrophages activate the IFN-β promoter upon MCMV infection [28,42–45]. Infection with MCMV m139*stop* resulted in significantly increased IFN-β promoter activation, suggesting that m139 dampens IFN-β induction upon MCMV infection (Fig 6A).

Next, we analyzed IFN-β transcription by qRT-PCR upon MCMV infection of iBMDM macrophages and SVEC4-10 endothelial cells, the two cell types in which MCMV m139*stop* showed a replication defect (Fig 2). As shown in Fig 6B, infection with MCMV m139*stop* resulted in significantly higher IFN-β transcription compared to cells infected with the m139-HA WT virus. By contrast, MCMV m139*stop* did not induce increased IFN-β transcription in SVEC4-10 endothelial cells (Fig 6C), suggesting that m139 curtails IFN-β induction in macrophages, but not in endothelial cells.

We next assessed the effect of m139 on the activation of type I IFN signaling in macrophages. Transcription of interferon-stimulated genes *Isg20* and *Cxcl10* was strongly increased

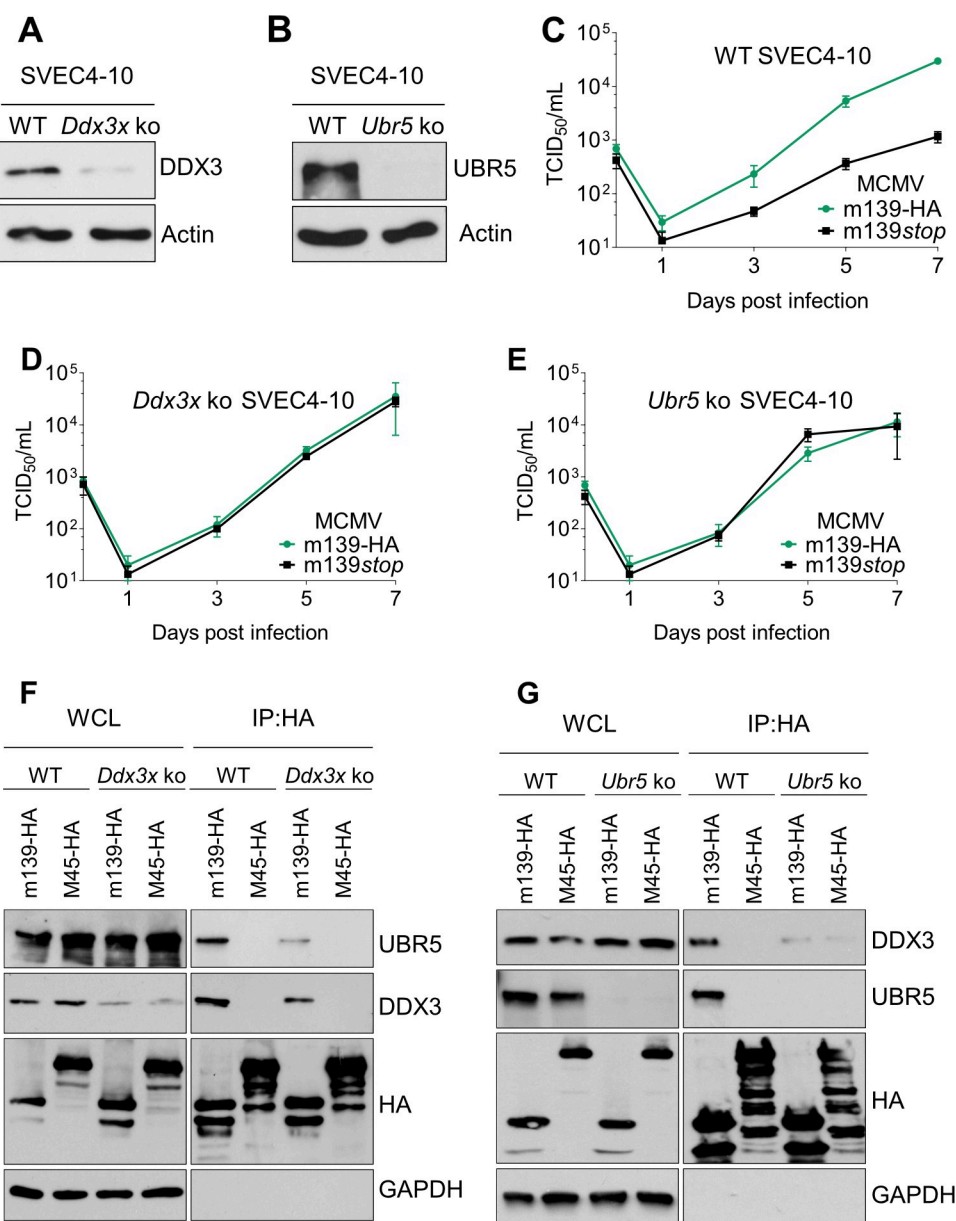

**Fig 5. Rescue of the MCMV m139*stop* replication defect in DDX3- and in UBR5-deficient SVEC4-10 cells.** (A, B) *Ddx3x* and *Ubr5* ko SVEC4-10 cells were generated by CRISPR/Cas9 gene editing. DDX3 (A) and UBR5 (B) expression was verified by immunoblot analysis. Note that the *Ddx3x* ko was incomplete. (C, D, E) For multistep replication kinetics, WT (C), *Ddx3x* (D), and *Ubr5* (E) ko SVEC4-10 cells were infected with MCMV m139-HA and MCMV m139*stop* (MOI = 0.01). At different days post infection, supernatants were collected for titration. Viral titers are shown as means ±SD of three biological replicates. (F, G) WT and *Ddx3x* ko (F) or *Ubr5* ko (G) SVEC4-10 cells were infected with MCMV m139-HA or MCMV M45-HA (control) at an MOI of 5 and harvested 24 hpi. The m139 protein was immunoprecipitated using an anti-HA affinity matrix. Co-precipitating proteins were detected by immunoblotting. The immunoblots shown in this figure are representative of two (F) or three (A, B, G) independent experiments.

in iBMDM infected with MCMV m139*stop* (Fig 6D and 6E). By contrast, the production of inflammatory cytokines IL-6 and TNF-α was only slightly increased in iBMDM infected with MCMV m139*stop* (Fig 6F and 6G).

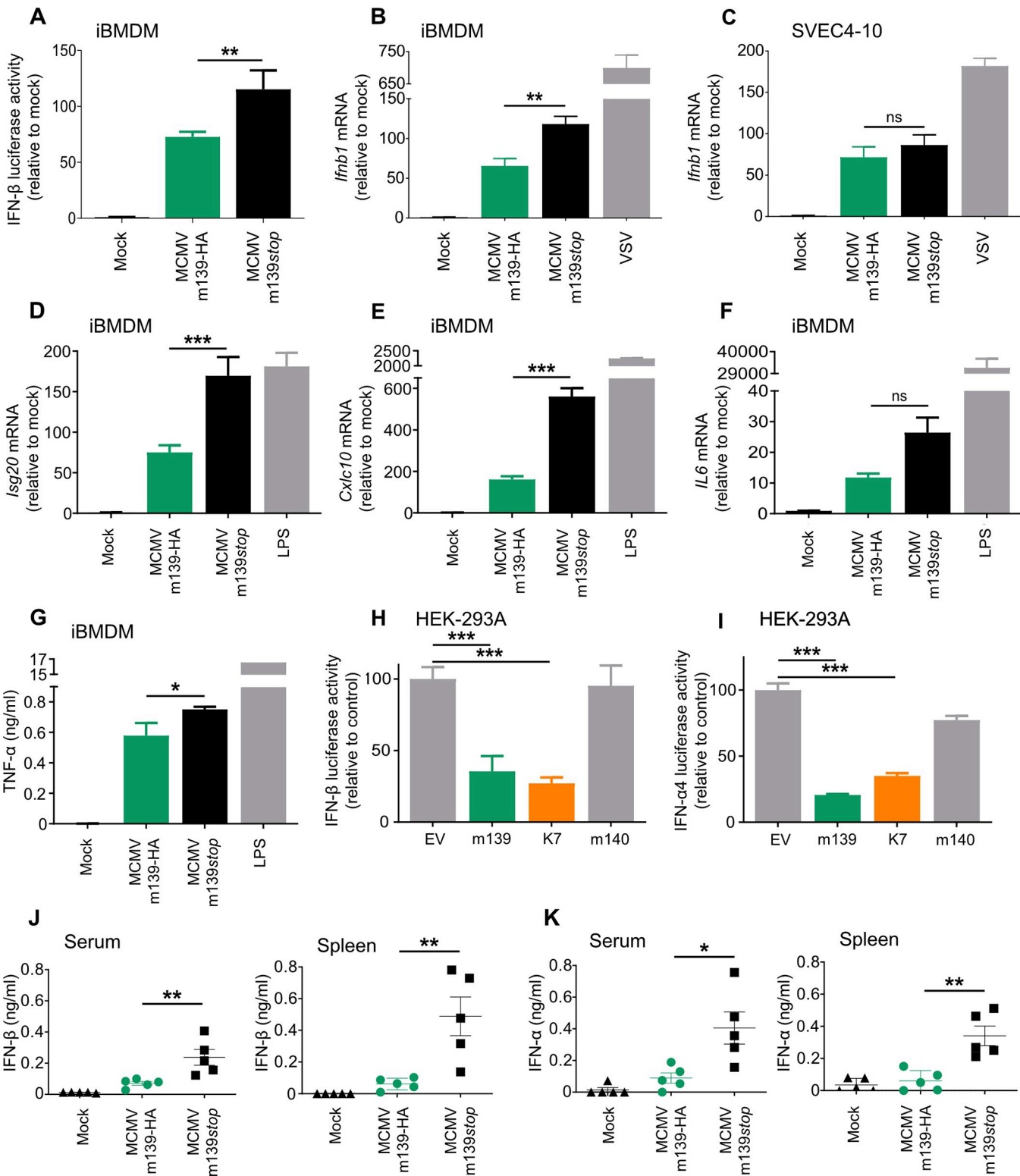

**Fig 6. MCMV m139 curtails type I IFN production.** (A) iBMDM macrophages expressing luciferase under the control of the IFN-β promoter were infected with MCMV m139-HA or m139*stop*. Luciferase activity was measured 8 hpi. Means ±SD of three biological replicates are shown. (B, C) iBMDM (B) or SVEC4-10 (C) cells were infected with MCMV m139-HA, MCMV m139*stop* or a recombinant vesicular stomatitis virus (VSV-GFP), a potent inducer of IFN-β transcription. RNA was harvested 6 hpi, and *Ifnb* transcripts were quantified by qRT-PCR. Means ±SD of three biological replicates are shown. (D, E, F) iBMDM were infected with MCMV m139-HA or MCMV m139*stop*. Cells treated with lipopolysaccharide (LPS, 100 ng/ml) were used as a positive control.

RNA harvested 6 hpi was used to quantify *Isg20* (D), *Cxcl10* (E), and *IL6* (F) transcripts by qRT-PCR. Means ±SD of three biological replicates are shown. (G) iBMDM were treated as above. Supernatants were collected 16 hpi to determine TNF-α levels by ELISA. Mean ±SD of three biological replicates are shown. (H, I) HEK-293A cells were co-transfected with DDX3 and IKKε (H) or DDX3 and IRF7 (I) expression plasmids, an IFNβ-luc (H) or an IFNα4-luc (I) reporter plasmid, and a renilla luciferase plasmid for normalization. Plasmids expressing MCMV m139, m140, VACV K7, or empty vector (EV) were co-transfected. Firefly and renilla luciferase activities were determined in the same samples. Values were normalized to those of cells co-transfected with EV. Means ±SD of three biological replicates are shown. (J, K) BALB/c mice were infected i.p. with $10^6$ PFU MCMV m139-HA or MCMV m139*stop*. IFN-β (J) and IFN-α (K) levels in serum and spleen were measured 8 hpi by ELISA. Means ±SEM are shown. Significance was determined by ANOVA. ns, not significant; *, p<0.05; **, p<0.01; ***, p<0.001.

The vaccinia virus (VACV) K7 protein has been shown to interact with DDX3 and block DDX3-mediated IFN-β induction by inhibiting IRF3 activation by the TBK1-IKKε complex [46]. To determine whether m139 inhibits DDX3-mediated IFN-β induction in a similar way as K7, we used a previously described luciferase reporter assay [46]. HEK-293A cells were transfected with DDX3 and IKKε expression plasmids to activate the IFN-β promoter by overexpression. Plasmids expressing m139, m140, or K7 were co-transfected, and IFN-β promoter activation was measured using a reporter plasmid expressing firefly luciferase under the control of the murine IFN-β promoter (IFNβ-Luc). In this assay, m139 and K7 strongly inhibited IFN-β promoter activation, whereas m140 did not (Fig 6H). As VACV K7 also inhibits DDX3-mediated IFN-α induction [47], we used a similar luciferase reporter assay to test whether m139 is able to inhibit IRF7-dependent activation of the IFN-α4 promoter. As shown in Fig 6I, m139 strongly inhibited activation of the IFN-α4 promoter. These results confirmed that m139 inhibits DDX3-dependent type I IFN induction. Interestingly, m139 also inhibited activation of the IFN-α4 promoter by a constitutively active IRF7 (S4 Fig), suggesting that m139 can inhibit IFN induction downstream of IRF7 activation, possibly by inhibiting DDX3 binding to IFN promoters [48].

To test whether m139 affects type I IFN induction in vivo, we infected mice intraperitoneally (i.p.) with MCMV and measured IFN-β and IFN-α levels in sera and in spleen homogenates by ELISA. At 8 hpi, significantly increased IFN-β and IFN-α levels were detected in mice infected with MCMV m139*stop* as compared to those infected with MCMV m139-HA (Fig 6J and 6K). These results demonstrate that m139 also dampens type I IFN production in vivo. Considering that MCMV m139*stop* has a replication defect in macrophages (Fig 2D), we wanted to determine whether this defect was dependent on DDX3 or UBR5. To do this, we generated DDX3 and UBR5-deficient iBMDM by CRISPR/Cas9 gene editing (Fig 7A and 7B). The previously observed replication defect of MCMV m139*stop* in WT iBMDM (Fig 7C) was rescued in DDX3-deficient macrophages (Fig 7D and S5A Fig). Interestingly, MCMV replication was strongly reduced in *Ubr5* knockout iBMDM (Fig 7E and S5B Fig), suggesting that UBR5 is important for efficient MCMV replication in these cells.

To determine whether the replication defect of MCMV m139*stop* in macrophages is dependent on DDX3-mediated IFN induction, we constructed a recombinant MCMV expressing VACV K7 in place of m139 (Fig 7F). K7 expression by the K7[Δm139] substitution mutant was confirmed by immunoblotting (Fig 7G). Next, we assessed the replication properties of the substitution mutant in iBMDM and SVEC4-10 cells. In iBMDM, K7 expression dampened IFN-α and IFN-β secretion (Fig 7H and 7I), and MCMV K7[Δm139] replicated to similar titers as the control virus (Fig 7J), indicating that the VACV K7 protein can rescue the replication defect of an m139-deficient virus in macrophages. By contrast, the K7[Δm139] substitution did not rescue the replication defect in SVEC4-10 endothelial cells (Fig 7K). This finding was not surprising as IFN-β transcription was not increased in the absence of m139 in SVEC4-10 cells (Fig 6C). Moreover, the replication defect of MCMV m139*stop* in SVEC4-10 cells depended on both DDX3 and UBR5 (Fig 5), which suggests that m139 antagonizes an IFN-independent function of DDX3 in endothelial cells.

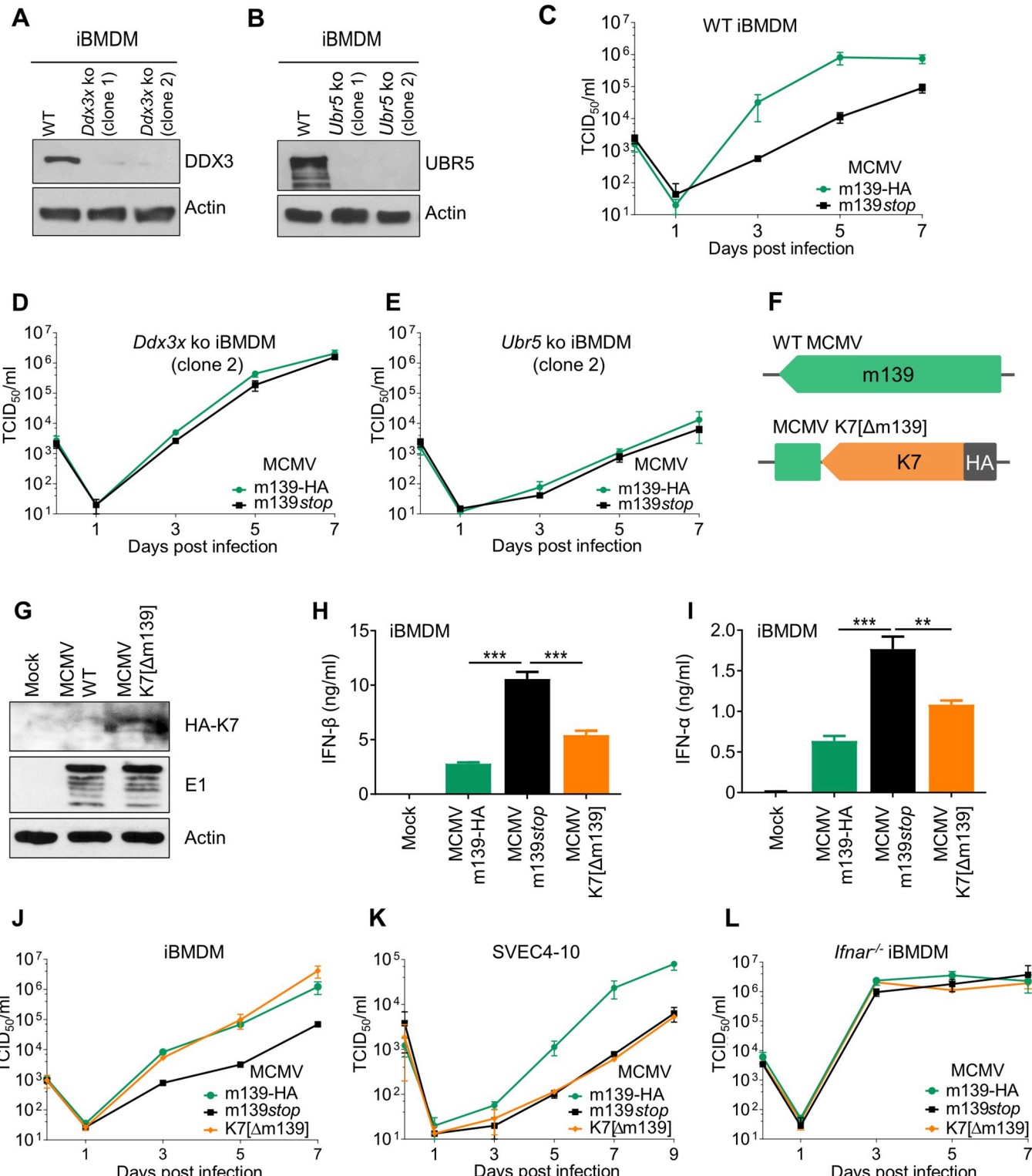

**Fig 7. Modulation DDX3-mediated IFN signaling by m139 is crucial for efficient replication in macrophages.** (A, B) *Ddx3x* and *Ubr5* ko iBMDM (two independent clones each) were generated by CRISPR/Cas9 gene editing. DDX3 (A) and UBR5 (B) expression was verified by immunoblot analysis. (C, D, E) For multistep replication kinetics, WT (C) *Ddx3x* (D) and *Ubr5* (E) ko iBMDM were infected with MCMV m139-HA or MCMV m139*stop* (MOI = 0.025). At different days post infection, supernatants were collected for titration. Viral titers are shown as means ±SD of three biological replicates. MCMV replication kinetics on the other *Ddx3x* and *Ubr5* ko cell clones are shown in S5 Fig. (F) Schematic representation of WT MCMV and MCMV K7[Δm139], a substitution

mutant expressing HA-tagged VACV K7 instead of m139. (G) 10.1 fibroblasts were infected with WT MCMV or K7[Δm139] (MOI = 5). Cell lysates were collected 24 hpi, and HA-K7 and E1 were detected by immunoblot analysis. (H, I) iBMDM were infected with MCMV m139-HA, m139*stop*, or K7[Δm139] (MOI = 0.1). At 16 hpi, IFN-β and IFN-α levels in the supernatant were measured by ELISA. Means ±SD of three biological replicates are shown. (J) For multistep replication kinetics, iBMDM were infected with MCMV m139-HA, m139*stop*, or K7[Δm139] (MOI = 0.025). Viral titers in the supernatant are shown as means ±SD of three biological replicates. (K, L) Multistep replication kinetics in (K) SVEC4-10 endothelial (MOI = 0.01) and (L) *Ifnar*$^{-/-}$ iBMDM (MOI = 0.05) with the same viruses as in J.

To further confirm that the replication defect of MCMV m139*stop* in iBMDM is IFN-dependent, we analyzed viral replication in *Ifnar*$^{-/-}$ iBMDM, which lack the type I IFN receptor. In these cells, MCMV m139-HA, m139stop, and K7[Δm139] replicated to the same high titers (Fig 7L). Taken together, the results shown in Fig 7 suggest that m139 inhibits DDX3-dependent type I IFN production in bone marrow-derived macrophages, and that this function is important for efficient MCMV replication in these cells.

## m139 is crucial for viral dissemination in vivo

To analyze the importance of m139 for MCMV dissemination in its host, we chose a route of infection that strongly depends on macrophages. MCMV administered by footpad-inoculation rapidly reaches the popliteal lymph nodes (PLN), where it infects subcapsular sinus macrophages. From there it spreads slowly to distant sites, suggesting that lymph node macrophages can be a bottleneck to viral dissemination [49]. We infected BALB/c mice with MCMV m139-HA, m139*stop*, or K7[Δm139] ($10^5$ PFU per mouse) and measured viral titers in PLN on day 3 post infection. Compared to the HA-tagged WT virus (MCMV m139-HA), infection with MCMV m139*stop* resulted in strongly reduced viral titers in PLN (Fig 8A). Expression of K7 by the K7[Δm139] substitution mutant did not restore viral titers in PLN to WT levels, but moderately improved titers compared to the m139*stop* mutant. Hence, m139 is required for efficient spread to or replication in PLN. Viral titers in the spleen (day 3 and day 7), and the liver (day 7) were below the detection limit, and viral titers in the lungs on day 7 post infection were low (Fig 8B). Such low titers are not unusual after footpad-infection with a moderate dose [49]. We also measured MCMV titers in the salivary glands, which are important organs for viral persistence and shedding [50]. In salivary glands, MCMV m139-HA was detected on day 7 and reached high titers by day 14 post infection (Fig 8C and 8D). Salivary gland titers of MCMV m139*stop* and K7[Δm139] MCMV were strongly reduced, which confirms the importance of m139 for viral dissemination *in vivo*. The substitution of m139 by K7 in the K7[Δm139] recombinant virus did not rescue the viral dissemination defect of the m139-deficient virus. Similar to MCMV m139*stop*, K7[Δm139] has a replication defect in endothelial cells (Fig 7K) and possibly in further cell types, suggesting that the VACV DDX3 antagonist K7 does not fulfill all functions of m139, particularly those related to UBR5. In summary, these findings show that m139 is required for efficient dissemination from a peripheral site of infection to local lymph nodes and to distant sites.

## Discussion

In this study, we show that MCMV m139 interacts with the host proteins DDX3 and UBR5 and inhibits DDX3-mediated type I IFN induction in macrophages. DDX3 is known to play an important role in antiviral innate immunity [40,41]. It participates in RNA sensing by RIG-I and related RNA helicases, interacts with the kinases TBK1 and IKKε to activate IRF3, and can bind directly to the IFN promoter [40,41]. More recently, DDX3 has also been shown to interact with IKKα and facilitate IRF7 activation by NIK and IKKα [47]. Two known sensors of MCMV infection, ZBP1/DAI and TLR9 [51–54], can induce type I IFNs through these

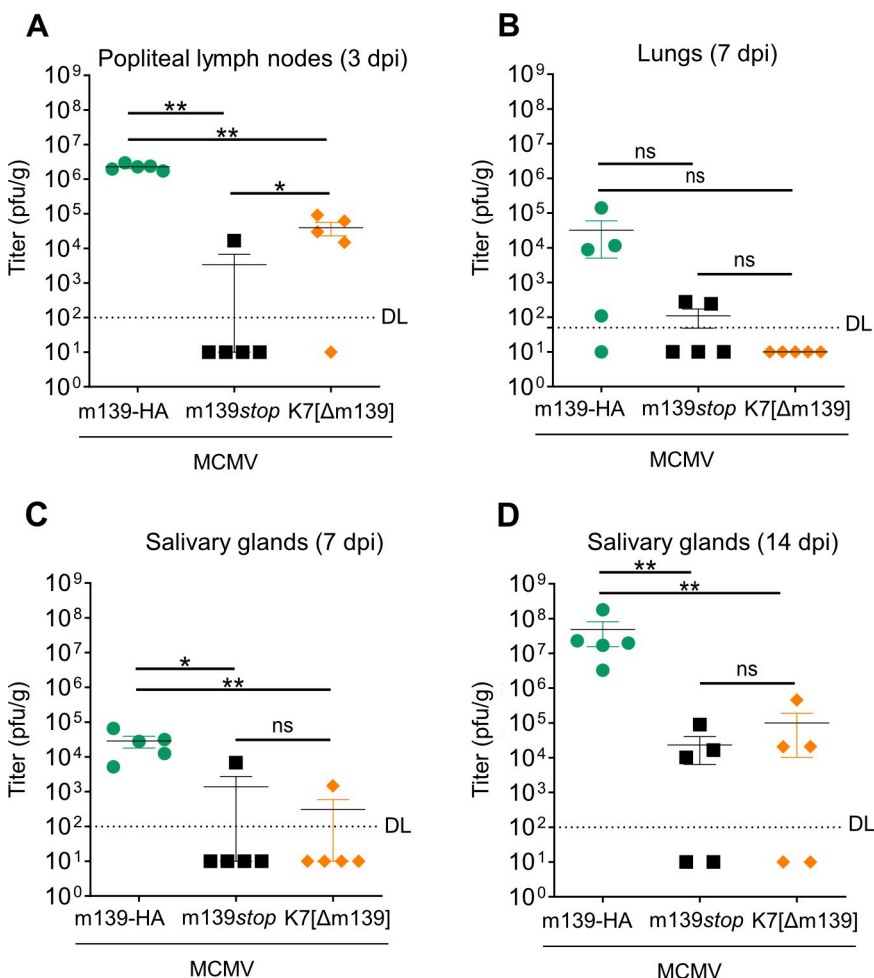

**Fig 8. m139 is crucial for MCMV dissemination.** BALB/c mice were infected by injection of $10^5$ PFU MCMV m139-HA, m139*stop*, or K7[Δm139] into the footpad. Mice (5 per group) were sacrificed on days 3, 7, and 14 post infection, and MCMV titers in different organs were determined by plaque assay. Titers in (A) popliteal lymph on day 3, (B) lungs on day 7, and salivary glands on day7 (C) and 14 (D) post infection are shown as mean titers ±SEM. DL, detection limit; ns, not significant; *, p<0.05; **, p<0.01.

kinase complexes [33,47]. Interestingly, IRF7 is expressed only in lymphocytes and myeloid cells, which comprise monocytes, macrophages, and dendritic cells [55]. This could be an explanation for the increased IFN-β transcription upon infection with MCMV m139*stop* in macrophages but not in endothelial cells (Fig 6B and 6C). Another possible reason could be the expression of the relevant pattern recognition receptors. TLR9, for instance, is expressed predominantly by antigen presenting cells such as macrophages and dendritic cells [56]. Only a few viral antagonists of DDX3-mediated IFN signaling have been described: The influenza virus protein PB1-F2 inhibits DDX3 by inducing its proteasomal degradation, while VACV protein K7 and hepatitis B virus polymerase block the interaction of DDX3 with IKKε [46,57,58]. This study describes the first DDX3 antagonist encoded by a herpesvirus, which inhibits IFN production in a similar fashion as VACV K7 (Fig 9A). Interestingly, there are two short stretches (5 to 8 amino acids) of similarity between m139 and K7, which might indicate a similar mode of interaction with DDX3. However, specific mutagenesis experiments would be required to verify the significance of these similarities.

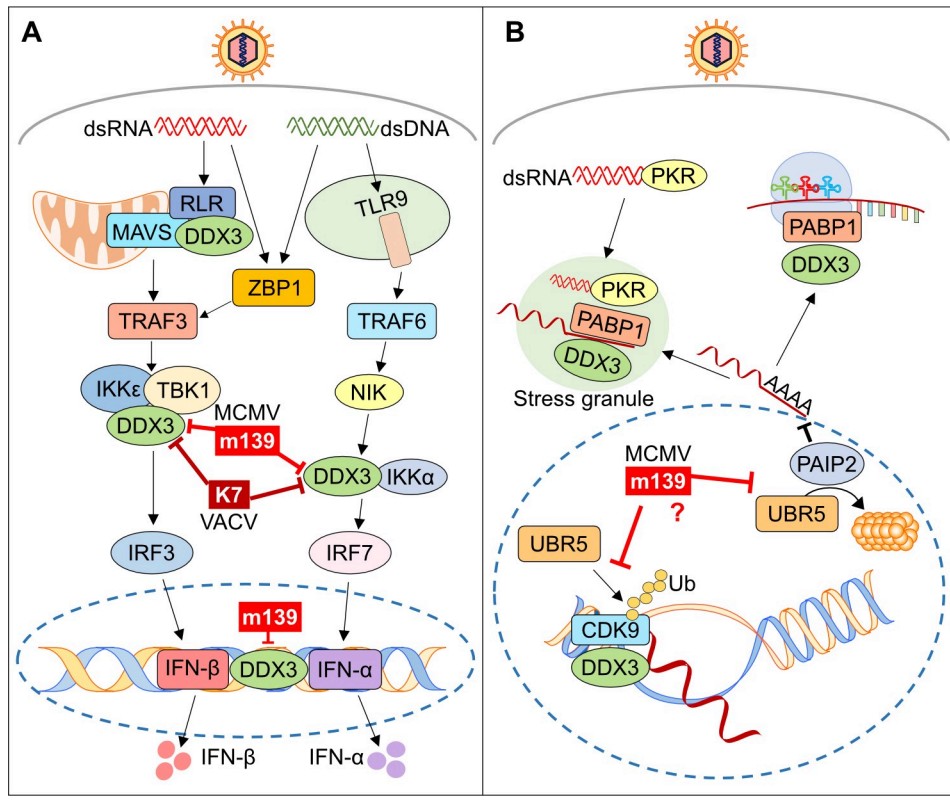

**Fig 9. Known functions of DDX3 and UBR5 and their presumed targeting by m139.** (A) Functions of DDX3 in IFN activating pathways. MCMV m139 interacts with DDX3 and inhibits IFN-β transcription either by targeting the interaction of DDX3 with IKKε (similar to VACV K7) or by blocking DDX3 binding to the IFN promoter; m139 also inhibits IFN-α induction in a similar way as K7. (B) Functions of DDX3 and UBR5 in RNA transcription, processing, and translation. MCMV m139 co-localizes with DDX3 and UBR5 in the nucleus, suggesting that m139 antagonizes DDX3 and UBR5-dependent functions in the nucleus.

Besides its role in innate immunity, DDX3 also participates in several aspects of RNA processing, including mRNA transcription, splicing, export, and translation initiation [59]. These functions affect a subset of cellular and viral RNAs containing specific structural features. For instance, DDX3 affects cell cycle regulation by facilitating nuclear export and translation of several cyclin mRNAs [59]. DDX3 also facilitates Rev-dependent nuclear export and translation of HIV mRNAs, demonstrating that viruses can exploit DDX3 for their own ends [40,59]. Although much less is known about the specific functions of UBR5, it is interesting to note that UBR5 is also involved in the regulation of transcription and translation [39]. While DDX3 promotes translation of specific mRNAs by interacting with the translation initiation factors eIF4A, eIF4G, and the polyadenylate-binding protein PABP1 [59], UBR5 supports this process by inducing the degradation of the PABP-interacting protein 2 (PAIP2), which functions as an inhibitor of PABP1 [60]. The fact that MCMV m139 interacts with both, DDX3 and UBR5 (Fig 3A–3C and S2 Fig), and co-localizes with both proteins in viral replication compartments (Fig 4) suggests that m139 might regulate the transcription, processing, or translation of specific viral or cellular mRNAs (Fig 9B) that affect the efficiency of viral replication in endothelial cells.

After footpad infection, MCMV mutants lacking m139 reached low titers in PLN on day 3 and in salivary glands on days 7 and 14 post infection (Fig 8). Viral dissemination to distant sites such as the salivary glands depends on the virus' ability to infect and persist in cells that

transport the virus as well as the virus' ability to replicate efficiently in target organs. In the case of MCMV, bone marrow-derived myelomonocytic cells are crucial for the transport of MCMV throughout the organism [3]. As m139-deficient MCMV mutants replicate poorly in macrophages, it seems likely that viral transport to distant sites is, at least in part, responsible for the observed dissemination defect. However, the experiments shown in this paper do not allow us to discriminate between impaired transport and reduced local replication. Thus, both could contribute to the dissemination defect.

The m139 gene is a member of the US22 gene family, which has 12 members in MCMV and 12 members in HCMV. Therefore, it would be interesting to know if m139 has a functional homolog in HCMV. Based on phylogenetic analysis, the closest sequence homologs are the HCMV genes US22, US23, US26. Little is known about the function of the corresponding proteins, but a proteomic study has identified the US22 protein as an RNA-binding protein [36]. The same study also found increased amounts of DDX3 bound to mRNAs of HCMV-infected cells, but it remains unknown whether or not these two observations are functionally connected. Another study has shown by siRNA-mediated knockdown that DDX3 acts as a proviral host factor during HCMV infection [35]. A recent mass spectrometry-based interactome analysis of all HCMV proteins did not identify DDX3 as a high-confidence interactor, but UBR5 was found to interact with six viral proteins, two of which are members of the US22 family: US26 and UL36 [61]. UBR5 also interacts with HCMV UL27 and its homologs in MCMV and rat CMV [62,63], but the consequences of these interactions remain unknown. Moreover, another study demonstrated a role of PAIP2, whose stability is regulated by UBR5/EDD1, as a restriction factor limiting productive HCMV infection [34].

Small animal models, such as the mouse model of CMV infection, help to verify the biological significance of identified virus-host interactions in an organism. Under ideal circumstances, the loss of a viral protein can be compensated by using a mouse strain in which the target protein has been removed by genetic knockout. This approach is promising if the viral protein only targets one host protein (e.g. [17,64]). In the present case, such a genetic rescue is unlikely to work as m139 interacts with at least two host proteins, DDX3 and UBR5. Nevertheless, a knockout of *Ddx3x* or *Ubr5* in SVEC4-10 endothelial cells was sufficient to restore the replication of the MCMV m139*stop* mutant to WT levels (Fig 5D and 5E), suggesting that DDX3 and UBR5 impinge on the same pathway in these cells. Moreover, a knockout of *Ddx3x* in bone marrow-derived macrophages rescued the replication of the MCMV m139*stop* mutant (Fig 7D and S5A Fig). Unfortunately, the same experiment cannot be done in mice as *Ddx3x* and *Ubr5* knockout mice are not viable [65,66]. One way to overcome this obstacle is the use of conditional knockouts such as the recently published conditional *Ddx3x* knockout mice [67]. Another approach consists of replacing the viral gene by another gene of known function, as we have done it by replacing MCMV m139 with a known DDX3 antagonist, VACV K7. Indeed, the substitution mutant rescued IFN-α and IFN-β secretion in macrophages and MCMV replication in BMDM in cell culture (Fig 7H–7J). However, the substitution only marginally improved viral replication in vivo (Fig 8A), probably because K7 cannot compensate for all functions of m139, particularly those related to UBR5.

Obviously, it would be highly interesting to know which DDX3 and UBR5-dependent functions m139 antagonizes in the nucleus of SVEC4-10 endothelial cells. Both, DDX3 and UBR5 affect the transcription and translation initiation of a subset of mRNAs (Fig 9B). Only a few mRNAs transcripts are known targets. However, the exact structural requirements for DDX3-mediated regulation and the complete set of transcripts affected are currently unknown. Additional research, which is beyond the scope of the present study, will be necessary to understand the details of transcriptional and translational regulation by DDX3 and UBR5 and to clarify how viral interference promotes viral replication.

## Materials & methods

### Ethics statement

Animal experiments were performed according to the recommendations and guidelines of the FELASA (Federation for Laboratory Animal Science Associations) and the Society of Laboratory Animals (GV-SOLAS) and approved by the institutional review board and local authorities (Behörde für Gesundheit und Verbraucherschutz, Amt für Verbraucherschutz, Freie und Hansestadt Hamburg, reference number N017/2019).

### In vivo experiments

Six to eight-week-old female BALB/c mice (Janvier Laboratories) were infected by subcutaneous injection of $10^5$ PFU of MCMV into the left footpad. Organs (PLN, spleen, liver, lungs, SG) were harvested on days 3, 7, and 14 post infection. Organ homogenates were titrated by plaque assay on M2-10B4 cells essentially as described [50]. For measurement of type I IFN, mice were infected by i.p. injection essentially as described [68]. Sera and spleens were harvested 8 hpi.

### Cell lines

SVEC4-10 (CRL-1658), NIH-3T3 (CRL-1658), TCMK-1 (CCL-139), M2-10B4 (CRL-1972), and HEK 293T (CL-11268) were obtained from the ATCC. HEK-293A cells (R705-07) were purchased from Invitrogen. Murine 10.1 fibroblasts have been described [69]. Immortalized murine bone marrow-derived macrophages (iBMDM) were obtained from BEI Resources, NIAID NIH (NR-9456). BMDM expressing firefly luciferase under control of the endogenous IFN-β promoter were isolated from transgenic mice [70] and immortalized by transduction with retrovirus J2 as described [71]. *Ifnar*$^{-/-}$ iBMDM have been described [45]. All cells were cultured at 37˚C and 5% $CO_2$ in complete Dulbecco's modified Eagle medium (DMEM) supplemented with 10% fetal calf serum and 100 IU penicillin/100 μg streptomycin.

### Viruses

The repaired WT MCMV Smith bacterial artificial chromosome (BAC), pSM3fr-MCK-2fl [72], was used to construct mutant MCMVs by *en passant* mutagenesis [73]. The m139-HA mutant was generated by inserting the hemagglutinin (HA) epitope sequence at the 3' end of the m139 ORF. m139*stop* was generated based on an HA-tagged mutant by introducing a point mutation at the position 195923 (T→A), which resulted in a stop codon at position 32 of m139. To generate MCMV K7[Δm139], the HA-K7 sequence was PCR-amplified from pCMV-HA-K7R and used to replace m139. To exclude unintended mutations, all recombinant BACs were sequenced at the NGS facility of the Heinrich Pette Institute. Infectious MCMV was reconstituted by transfection of 10.1 cells with BAC DNA. Virus stocks were titrated using the median tissue culture infective dose (TCID$_{50}$) method. Centrifugal enhancement ($1000 \times g$, 30 min) was applied for high-MOI infections. VSV-GFP [74] was kindly provided by César Muñoz-Fontela (Bernhard Nocht Institute for Tropical Medicine, Hamburg, Germany).

### Growth curves

Multistep replication kinetics were done as described [69]. Cells were seeded and infected in 6-well dishes. Three hpi cells were washed twice with phosphate-buffered saline (PBS), and fresh medium was added. Supernatants harvested from infected cells were titrated on 10.1 cells.

## Plasmids

pcDNA3-HA-m139, pcDNA3-HA-m140, and pcDNA-m141-HA were generated by PCR-amplification of the respective ORF from the MCMV BAC with primers containing restriction sites and an HA tag sequence. The PCR products were cloned in pcDNA3 (Invitrogen) using EcoRI and EcoRV restriction sites for HA-m139, HindIII and XbaI for HA-m140, and HindIII and XbaI for m141-HA. pCMV-HA-K7R [46] was provided by Martina Schröder (Maynooth University, Ireland). pcDNA3-IKKε, encoding human I-kappa-B kinase epsilon [75], was provided by Rongtuan Lin (McGill University, Montreal, Canada). pcDNA-HA-DDX3 was obtained from Addgene (plasmid 44975). pCAGGS-IRF7, pCAGGS-IRF(2D), and pGL3-IFNα4-luc [76] were provided by Georg Kochs (University of Freiburg, Germany). pGL3basic-IFNβ-Luc (IFNβ-Luc) has been described [45]. pRL-TK was purchased from Promega. NIH-3T3 fibroblasts were transfected using Lipofectamine (Thermo Fisher Scientific) according to the manufacturer's instructions.

## Generation of knockout cells

The lentiviral CRISPR/Cas9 vector pSicoR-CRISPR-PuroR was used was used to generate *Ddx3x* and *Ubr5* knockout iBMDM and SVEC4-10 cells essentially as previously described [77]. Guide RNAs targeting *Ubr5* (GGCTACTATTAAACAGTGTG and GCTCCAGTACA TTCAGAGGT) and *Ddx3x* (CTATCTTTACTAGAACTCCA) were designed using E-CRISP (http://www.e-crisp.org/E-CRISP). Lentivirus production, transduction of iBMDM or SVEC4-10 cells, and selection of clones was done as previously described [77]. Cell clones were isolated by limiting dilution and screened by immunoblot analysis.

## Antibodies and reagents

Monoclonal antibodies against HA (3F10; Roche), β-actin (AC-74; Sigma), glyceraldehyde-3-phosphate dehydrogenase (GAPDH) (14C10; Cell Signaling), DDX3 (C-4; Santa Cruz Biotechnology), UBR5 (B-11; Santa Cruz Biotechnology) and polyclonal antibodies against LSD1 (Cell Signaling) were obtained from commercial sources. Antibodies against MCMV IE1 (CROMA101) and M112-113 (CROMA103) were provided by Stipan Jonjic (University of Rijeka, Rijeka, Croatia). Secondary antibodies coupled to horseradish peroxidase (HRP) were purchased from DakoCytomation or Jackson ImmunoResearch. Secondary antibodies coupled to Alexa-488, Alexa-555 or Alexa-633 were purchased from Thermo Fisher Scientific. Lipopolysaccharide (LPS) was purchased from Invivogen.

## Immunofluorescence

Cells were seeded on 8-well μ-slides (Ibidi) one day before transfection or infection. On the following day, cells were transfected with expression plasmids using lipofectamine transfection reagent (Thermo Fisher Scientific), or infected at an MOI of 1 $TCID_{50}$/cell. At defined time points, cells were fixed using 4% paraformaldehyde for 15 minutes at RT. The remaining aldehyde groups were blocked using 50 mM ammonium chloride. Cells were permeabilized with 0.3% Triton X-100 for 15 min and blocked with 0.2% gelatin (Sigma). Antibodies were diluted in 0.2% gelatin, applied to the cells, and incubated for 1 hour. Hoechst 33342 (Thermo Fisher Scientific) was used to stain nuclear DNA. Fluorescence images were acquired with a Nikon A1 confocal laser scanning microscope.

## Luciferase reporter assay

One day before infection, $3 \times 10^5$ iBMDM β-luc reporter macrophages were seeded in 12-well plates. Cells were infected at an MOI of 3 $TCID_{50}$/cell, washed 3 hpi with PBS, and incubated

with fresh growth medium. 8 hpi cells were lysed in Cell Culture Lysis Reagent (Promega). Lysates were combined with Luciferase Substrate (Promega) and luminescence was measured with a FLUOstar Omega plate reader (BMG Labtech). For the IFN-β luciferase assay, $1.5 \times 10^5$ HEK-293A cells were seeded in 12-wells plates and transfected using Lipofectamine 2000 (Thermo Fisher Scientific) with 180 ng pcDNA3-DDX3-HA, 180 ng pcDNA3-IKKε, 300 ng pGL3basic-IFNβ-Luc, 30 ng pRL-Renilla, and 500 ng of a viral protein expression plasmid or an empty pcDNA3 vector. For the IFN-α4 luciferase assay, $2 \times 10^5$ HEK-293A cells were seeded in 12-wells plates and transfected with 400 ng pcDNA3-DDX3-HA, 400 ng pCAGG-S-IRF7 or pCAGGS-IRF(2D), 500 ng pGL3basic-IFNα4-Luc, 50 ng pRL-Renilla, and 500 ng of a viral protein expression plasmid or an empty pcDNA3 vector. 24 hours post transfection, cells were lysed, and firefly and renilla luciferase activities were determined using a Dual Luciferase Assay (Promega) and a FLUOstar Omega plate reader (BMG Labtech).

## Cell fractionation

Nuclear and cytoplasmic fractions of MCMV-infected 10.1 cells ($2 \times 10^6$ cells/well) were lysed stepwise using an NE-PER nuclear and cytoplasmic extraction kit (Thermo Fisher Scientific) according to the manufacturer's instructions. Protein concentrations were determined using a BCA assay (Thermo Fisher Scientific). Equal quantities of nuclear and cytoplasmic proteins were separated by SDS-PAGE and analyzed by immunoblotting

## Immunoprecipitation and immunoblotting

For immunoprecipitation, cells were seeded in 6-well plates one day before infection. The following day the cells were infected at an MOI of 3 $TCID_{50}$/cell. 24 hpi cells lysed using NP-40 buffer (50 mM Tris, 150 mM NaCl, 1% Nonidet P-40, and Complete Mini protease inhibitor cocktail [Roche]). Insoluble material was removed by centrifugation. The remaining supernatant was precleared with protein G Sepharose (PGS, GE Healthcare). m139-HA was immunoprecipitated with anti-HA affinity matrix (clone 3F10, Roche). DDX3 and UBR5 were precipitated with specific antibodies and PGS beads. Precipitates were washed 3 times with buffer 1 (1 mM Tris pH 7.6, 150 mM NaCl, 2 mM EDTA, 0.2% NP-40), twice with buffer 2 (1 mM Tris pH 7.6, 500 mM NaCl, 2mM EDTA, 0.2% NP-40) and once with buffer 3 (10mM Tris pH 7.6). They were then eluted by boiling in SDS-PAGE sample buffer (125 mM Tris pH 6.8, 4% SDS, 20% glycerol, 10% β-mercaptoethanol, 0.002% bromophenol blue) and subjected to SDS-PAGE and immunoblot analysis.

For immunoblot analysis, SDS-PAGE sample buffer or NP-40 buffer were used. Equal amounts of protein (NP-40) or lysate (SDS-PAGE sample buffer) were separated by SDS-PAGE and subsequently transferred to a nitrocellulose (Amersham) or 0.2 μm polyvinylidene difluoride (Immobilon-PSQ, Sigma) by semi-dry or wet blotting, respectively. Proteins of interest were detected with protein-specific primary antibodies and HRP-coupled secondary antibodies by enhanced chemiluminescence (GE Healthcare) and imaged using X-ray films or a Fusion Capture Advance FX7 16.15 (Peqlab) camera.

## SILAC, AP-MS, and LC-MS/MS

For SILAC, SVEC4-10 cells were grown for 8 passages in SILAC medium, supplemented with 10% dialyzed FCS, 4 mM glutamine, 100 IU penicillin/100 μg streptomycin in the presence of either $^{13}C_6,^{15}N_2$-lysine / $^{13}C_6,^{15}N_4$-arginine (SILAC heavy) or unlabeled lysine / arginine (SILAC light). Heavy and light labeled SVEC4-10 cells were infected with MCMV m139-HA and MCMV WT (MOI = 5) and lysed 24 hpi in sterile-filtered NP-40 buffer. 1 mg of each whole cell lysate was used for immunoprecipitation with an anti-HA affinity matrix.

Immunoprecipitates were washed six times with sterile-filtered minimal washing buffer (50 mM Tris, 150 mM NaCl, 10% (v/v) glycerol, pH 7.5) and eluted by boiling in elution buffer (1% (w/v) SDS in 50 mM Tris, 150 mM NaCl, pH 7.5). The eluted samples from heavy and light-labeled cells were mixed in a 1:1 ratio, separated by a short (1 cm) SDS-PAGE run to remove interfering substances, and stained with Coomassie R-250. Each lane was cut into 1 $mm^3$ cubes followed by destaining. Disulfide bonds of proteins in the gel matrix were reduced in presence of 10 mM dithiotreitol (Fluka), alkylated in presence of 20 mM iodoacetamide (IAA, Sigma), and digested with trypsin (50:1 protein:enzyme ratio, Promega) overnight. Peptides were eluted from the gel pieces and dried in a vacuum concentrator. Samples were resuspended in 0.1% formic acid (FA) and transferred into a full recovery autosampler vial (Waters). Chromatographic separations were achieved on a Dionex Ultimate 3000 UPLC system (Thermo Fisher Scientific) with a two-buffer system (buffer A: 0.1% FA in water, buffer B: 0.1% FA in acetonitrile (ACN)). Attached to the UPLC was a C18 trapping column (Acclaim PepMap 100, 100 μm × 2 cm, 100 Å pore size, 5 μm particle size) for desalting and purification followed by a C18 analytical column (Acclaim PepMap 100, 75 μm × 50 cm, 100 Å pore size, 2 μm particle size). Peptides were separation using a 110 min gradient with increasing ACN concentration from 2 to 32% ACN. The eluting peptides were analyzed on a tribrid Quadrupole Iontrap Orbitrap mass spectrometer (Fusion, Thermo Fisher Scientific) in data-dependent acquisition (DDA) mode. For DDA, the mass spectrometer was operated in Orbitrap–Iontrap mode at top speed for precursor selection for fragmentation. Therefore, observed precursors with charge stages +2 to +5 in a range from 400 to 1300 m/z in a MS1 survey scan ($2 \times 10^5$ ions, 120,000 resolution, 120 ms fill time) were analyzed by MS/MS (HCD at 30 normalized collision energy, $1 \times 10^4$ ions, 60 ms fill time) within 3 s. A dynamic precursor exclusion of 20 s was used.

## Data analysis and processing

Acquired DDA LC-MS/MS data were searched against the mouse SwissProt protein database downloaded from Uniprot (release August 2017, 16,909 protein entries) and an MCMV protein library using the Sequest algorithm integrated in the Proteome Discoverer software version 2.0. Mass tolerances for precursors was set to 10 ppm and 0.6 Da for fragments. Carbamidomethylation was set as a fixed modification for cysteine residues and $^{13}C_6, ^{15}N_2$ lysine, $^{13}C_6, ^{15}N_4$ arginine, the oxidation of methionine, pyro-glutamate formation at glutamine residues at the peptide N-terminus as well as acetylation of the protein N-terminus, methionine loss at the protein N-terminus and the acetylation after methionine loss at the protein N-terminus were allowed as variable modifications. Only peptide with a high confidence (false discovery rate < 1% using a decoy data base approach) were accepted as identified. Proteome Discoverer search results were imported into Skyline software version 4.2 allowing only high confidence peptides. Precursor traces (M, M+1, M+2) were extracted. Precursors with an idot product of >0.9 in at least one sample were kept. For each peptide, the ration of heavy to light was calculated and the median peptide ratio per protein was estimated, which were then used for relative comparison. Amino acid sequence analysis of m139 and NES prediction was done using the computational tool Wregex [78].

## Quantitative RT-PCR

To quantify IFN-β transcripts, cells were infected at a MOI of 0.1 $TCID_{50}$/cell. Three hpi cells were washed with PBS and fresh medium was added. Extraction of total RNA, cDNA synthesis and quantitative PCR (qPCR) was performed as described [77]. The following primers were used for amplification of transcripts: *Ifnb1* (CTGGCTTCCATCATGAACAA and

AGAGGGCTGTGGTGGAGAA), *Cxcl10* (CTGCTGGGTCTGAGTGGGACT and CCTATGGCCCTCATTCTCACTG), *Isg20* (GAACATCCAGAACAACTGGCG and GTA GAGCTCCATTGTGGCCCT), *IL6* (GCTACCAAACTGGATATAATCAGGA and CCA GGTAGCTATGGTACTCCAGAA) and *Actb* (AGAGGGAAATCGTGCGTGAC and CAA TAGTGATGACCTGGCCGT). Gene transcripts were quantified using the ΔΔCt method and normalized to the housekeeping gene *Actb*.

## ELISA

IFN-β production were measured using the LumiKine mouse IFN-β ELISA kit (Invivogen) according to the manufacturer's instructions. IFN-α production was detected using an IFN-α capture antibody (RMMA-1, PBL Assay Science) and a rabbit anti-mouse IFN-α detection antibody (#32100–1, PBL Assay Science). TNF-α production was detected using an anti-mouse TNF capture antibody (TN3-19.12, BD Biosciences) and human anti-mouse TNF (516D1A1, BD Biosciences). For measurement of IFN-β and IFN-α production, iBMDM were infected at MOI 0.1, for measurement of TNF-α with MOI 1. Supernatants were collected 16 hpi. Type I IFN levels in the serum and the spleen of mice were determined as previously described [45].

## Statistical analysis

Statistical analyses were performed using GraphPad Prism 5.0 Software. One-way ANOVA with Bonferroni post hoc test was used for the analysis of qRT-PCR and luciferase reporter assays. Statistical significance of in vivo experiments was assessed using the Mann-Whitney test.

## Supporting information

**S1 Fig. The m139 protein is recruited to viral replication compartments by E1.** NIH-3T3 fibroblasts were transfected with a plasmid encoding HA-tagged MCMV m139 (A) or plasmids encoding m139 and the MCMV E1 proteins (B). Cells were fixed 24 hpi and analyzed by immunofluorescence using antibodies specific for the HA epitope tag and E1. Nuclei were stained using Hoechst 33342. Images were obtained by confocal laser scanning microscopy and are representative of three independent experiments. Scale bar, 10 μm. (TIFF)

**S2 Fig. m139 interacts with DDX3 and UBR5 in iBMDM.** iBMDM were infected with MCMV m139-HA or M45-HA at an MOI of 5. Cell lysates were collected 24 hpi and subjected to immunoprecipitation (IP) using an anti-HA affinity matrix. Co-precipitating proteins were detected by immunoblotting with specific antibodies. (TIFF)

**S3 Fig. DDX3 levels are not decreased in MCMV-infected cells.** (A) SVEC4-10 cells and (B) iBMDM were infected with MCMV m139-HA or MCMV m139*stop* at an MOI of 5. Whole cell lysates were prepared at the indicated times post infection and analyzed by immunoblot analysis. (TIFF)

**S4 Fig. m139 inhibits IFN-α4 promoter activation downstream of IRF7 activation.** HEK-293A cells were co-transfected with DDX3 and IRF7(2D) expression plasmids, an IFNα4-luc reporter plasmid, a renilla luciferase normalization control. Plasmids expressing MCMV m139, m140, VACV K7, or empty vector (EV) were co-transfected. IRF7(2D) is a

constitutively active IRF7. Firefly and renilla luciferase activities were determined in the same samples. Values were normalized to those of cells co-transfected with EV. Means ±SD of three biological replicates are shown. The result is representative of three independent experiments. (TIFF)

**S5 Fig. Replication of MCMV m139stop in DDX3 and UBR5-deficient macrophages.** (A) *Ddx3x* ko iBMDM (clone 1) or (B) *Ubr5* ko iBMDM (clone 1) were infected with MCMV m139-HA or m139*stop* (MOI = 0.025). Virus release into the supernatant was quantified by titration. Viral titers are shown as means ±SD of three biological replicates. (TIFF)

## Acknowledgments

The authors would like to thank Martina Schröder, Rongtuan Lin, Georg Kochs, César Muñoz-Fontela, and Stipan Jonjic for providing materials, the NGS facility of the Heinrich Pette Institute (Daniela Indenbirken, Malik Alawi, and Sanamjeet Virdi) for sequencing, and Markus Stempel and Chris Benedict for helpful advice.

## Author Contributions

**Conceptualization:** Olha Puhach, Wolfram Brune.

**Formal analysis:** Olha Puhach, Eleonore Ostermann, Christoph Krisp, Hartmut Schlüter, Wolfram Brune.

**Funding acquisition:** Wolfram Brune.

**Investigation:** Olha Puhach, Eleonore Ostermann, Christoph Krisp.

**Methodology:** Christoph Krisp, Hartmut Schlüter, Melanie M. Brinkmann.

**Resources:** Melanie M. Brinkmann.

**Supervision:** Giada Frascaroli, Hartmut Schlüter, Melanie M. Brinkmann, Wolfram Brune.

**Writing – original draft:** Olha Puhach, Wolfram Brune.

**Writing – review & editing:** Olha Puhach, Eleonore Ostermann, Christoph Krisp, Giada Frascaroli, Hartmut Schlüter, Melanie M. Brinkmann, Wolfram Brune.

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
