## [Decision Letter · Decision Letter 0]

31 May 2020

Dear Professor Brune,

Thank you very much for submitting your manuscript "Murine cytomegaloviruses m139 targets DDX3 to curtail interferon production and promote viral replication" for consideration at PLOS Pathogens. As with all papers reviewed by the journal, your manuscript was reviewed by members of the editorial board and by several independent reviewers. In light of the reviews (below this email), we would like to invite the resubmission of a significantly-revised version that takes into account the reviewers' comments.

The three reviewers were consistent in their enthusiasm for your manuscript. However, there were also three common concerns that are evident in their reviews that would certainly need addressing in a revised manuscript. These were: 1) the need to further solidify the link between m139 and IFN expression, 2) required consolidation of the in vivo data (also with investigation of the relationship between m139 and IFN in vivo) and 3) the utility of UBR5 and/or DDX3 deficient cells in m139-stop rescue experiments to further solidify the relationship between these viral and host proteins.

We cannot make any decision about publication until we have seen the revised manuscript and your response to the reviewers' comments. Your revised manuscript is also likely to be sent to reviewers for further evaluation.

Sincerely,

Ian R Humphreys

Guest Editor

PLOS Pathogens

Klaus Früh

Section Editor

PLOS Pathogens

Kasturi Haldar

Editor-in-Chief

PLOS Pathogens

orcid.org/0000-0001-5065-158X

Michael Malim

Editor-in-Chief

PLOS Pathogens

orcid.org/0000-0002-7699-2064

The three reviewers were consistent in their enthusiasm for your manuscript. However, there were also three common concerns that are evident in their reviews that would certainly need addressing in a revised manuscript, namely 1) further solidifying the link between m139 and IFN expression, 2) consolidation of the in vivo data (an early examination of IFN and virus replication would be useful to disentangle effects on possible IFN-mediated control of early virus replication versus an unrelated dissemination phenotype) and 3) utility of UBR5 and/or DDX3 deficient cells in m139-stop rescue experiments to further solidify the relationship between these viral and host proteins.

Reviewer's Responses to Questions

**Part I - Summary**

Reviewer #1: The manuscript by Puhach and co-workers extends studies on the murine cytomegalovirus (MCMV) protein m139, a member of the US22 gene family previously shown to be required for efficient viral replication in macrophages, and to be a potential inhibitor of IFN-beta induction. Here, the authors provide mechanistic insights into how m139 achieves that. They report that this viral protein interacts with the host DEAD box helicase DDX3 and with the E3 ubiquitin ligase UBR5 to promote viral replication in macrophages and endothelial cells. They demonstrate that via DDX3, the m139 protein inhibits IFN-beta production in macrophages, while in endothelial cells the increased replication associated to DDX3 and UBR5 is not due to a blockade of IFN-beta induction. Finally, they describe that, in vivo, m139 is key to viral dissemination to local lymph nodes and to the salivary glands. The manuscript is properly organized, well executed, and reports interesting results. While virally-encoded DDX3 inhibitors that dampen interferon production have been already described, m139 represents the first example found in herpesviruses. Thus, the study describes an important additional strategy of host innate immune evasion used by MCMV.

However, there are some relevant issues that are not addressed in the manuscript at present. Does m139 inhibit IFN-beta production in vivo? The interaction of m139 with DDX3/UBR5 probably leads to the alteration of additional cellular targets besides IFN-beta This is an important aspect that needs to be determined.

Reviewer #2: The manuscript by Puhach et al. deals with a novel molecule involved in the innate immunity against herpesviruses, and the respective antagonism by cytomegaloviruses.

It uncovers important parallels to poxviruses and their Bcl2-like innate immune evasins.

The topic is highly relevant. The paper is nicely written and clear.

Reviewer #3: Puhach et al. addressed the function of the murine cytomegalovirus open reading frame m139. To this end the authors generated a recombinant virus expressing HA-tagged m139. They detected early kinetics of m139 expression. Obviously the viral E1 protein recruits m139 to pre-replication compartments in the nucleus. Studies with cell lines revealed that m139 is required for efficient MCMV replication in macrophages and endothelial cells, whereas it is dispensable in fibroblasts and epithelial cells. m139 interacts with the viral m140 and m141 proteins, but not E1, and with the host components DDX3, UBR5 and IFIT1. Replication of m139stop virus was rescued in DDX3 as well as UBR5-deficient cells suggesting that DDX3 and UBR5 are restriction factors of MCMV replication, which are counteracted by m139. Furthermore, the interaction of m139 with DDX3 seems to depend on UBR5 and vice versa. Finally, m139 inhibits DDX3-dependent IFN-beta induction and m139 is required efficiently infect distal sites such as popliteal lymph nodes and salivary glands.

The authors have chosen a straight forward approach to study the function of m139, generation of recombinant viruses either expressing HA-tagged m139 or m139stop. This is a solid study that is based on overall well controlled experiments. The data are interesting for virologists specialized on MCMV. The presented data further illuminate the complex interactions of viral components with host factors.

**Part II – Major Issues: Key Experiments Required for Acceptance**

Reviewer #1: 1.Since DDX3 and UBR5 are enzymes involved in numerous host processes, participating in the regulation at different levels of cellular mRNAs, most likely the effects of the interaction of m139 with these two host components are not limited to the inhibition of IFN-beta. Taking this in consideration, an analysis of the m139 effects on selected DDX3/UBR5 targets should be performed. Is IFN-alpha production also altered by m139? Anti-inflammatory cytokines? Related to this issue: the authors nicely show that the replication defect of MCMV m139 stop is rescued in DDX3- and in UBR5-deficient cells. Following this same line, can the replication defect of the mutant virus be reverted in IFNalphaRko iBMDMs? If this is the case, in which extent?

2. The authors demonstrate that the m139-deficient MCMV is compromised in its ability to replicate in vivo in popliteal lymph nodes and salivary glands. However, they haven’t formally proven that the decreased in vivo growth of the mutant is due, at least in part, to an increase in IFN-beta levels. At a minimum, the levels of IFN-alpha and IFN-beta in serum and selected organs of mice infected with the m139-deficient MCMV should be analyzed and compared to those in MCMV infected mice under the same conditions. In addition, can the decreased replication of MCMV m139 stop observed in popliteal lymph nodes and salivary glands be extended to other key organs?

3. The authors haven’t aimed to identify the specific DDX3-dependent cellular process/es interfered by m139. The manuscript would benefit from the inclusion of assays addressing this issue, for example to discern whether the viral protein targets the interaction of DDX3 with IKK-epsilon or blocks DDX3 binding to the IFN-beta promoter.

Reviewer #2: The manuscript follows two aspects:

(I) In an endothelial cell line (SVEC4-10), an interaction of the MCMV-encoded m139 with UBR5 and DDX3X as well as an UBR5- and DDX3X-dependent growth impairment of the m139-deficient MCMV mutant was uncovered.

(II) In the macrophage-like cell line NR‐9456, m139 was found to be a functional analogue of (and to be functionally replaceable by) the VACV-encoded K7 in terms of IFNβ induction.

Given that K7 targets DDX3X in order to prevent IFNβ induction, both lines of evidence appear to converge nicely.

However, the m139 coding capacity seems not affect the IFNβ induction in SVEC cells (although there seems be a trend in this direction) and the K7 insertion does not revert the replication of Δm139-MCMV in SVEC cells.

Thus, the functional association of the m139-DDX3X interaction and the effect on IFNβ induction - as stated in the title - is so far not sufficiently substantiated by experimental evidence.

Key experiments are:

(I) To test if m139-HA precipitates UBR5, DDX3X, and IFIT1 in NR‐9456 cells.

(II) To test if an UBR5 and/or DDX3X knock-out /-down reverts the Δm139-MCMV replication in NR‐9456 cells.

Honestly, I would drastically increase the number of replicates in Fig. 6C. If Δm139-MCMV shows an increased IFNβ induction there, most of the discrepancy vanishes.

Given that K7 binds a hell of proteins (PMID: 28815417), I would also check how Δm139-MCMV:K7 replicates in the SVEC clones either lacking UBR5 or DDX3X just to rule out that K7 causes some weird problems in this respect.

SVEC cells are SV40-transformed cells.

Couldn’t the authors assess Δm139-MCMV induction and replication in primary mouse ECs (e.g., PMID: 28060318) or Luka’s EC model (PMID: 23773211)?

-Honestly, the in vivo analysis is a bit premature. Some more organs and time points should be assessed.

Reviewer #3: Usually, VSV is a very potent inhibitor of IFN responses. Thus, the statement in line 232/233 could be misleading. Please indicate which VSV strain was used.

The control experiment that replication of MCMV m139stop can be rescued in DDX3 ko or UBR5 ko iBMDM should be included.

The conclusions in line 13/14 and line 283/284 are formulated too strong. Since M139 is required for efficient replication at local sites it is well possible that the reduced local virus titers are not sufficiently high in order to support virus transport to secondary target organs, especially the salivary glands. This would results in diminished MCMV titers in the salivary glands as shown in Fig. 8. Thus, if no other experiments are presented it remains unclear whether reduced local replication of MCMV m139stop or reduced dissemination is the underlying mechanism. Importantly, in this study it was not addressed whether m139 deficiency somehow affects the transporting myeloid cells itself.

**Part III – Minor Issues: Editorial and Data Presentation Modifications**

Reviewer #1: Minor points:

1. Fig. 6A should be quoted in the text. Eliminate “Error! Reference source not found” in l. 229.

2. The number of independent experiments performed need to be incorporated in some Figure legends (e.g. Figure 8).

3. Black and white images in some of the panels in Figures 1C and 4 could be provided in color for a better visualization.

4. Table 1, a legend to describe L/H H/L should be included.

Reviewer #2: To line 227: IFNβ induction by MCMV was previously shown for a macrophage cell line by Le et al. (PMID: 18420790) and for primary macrophages by Döring et al. and Ehlting et al. (PMIDs: 25231302 & 26299622). These papers should be cited.

Interactions of CMV-encoded proteins with UBR5 have been shown before (PMID: 30619335, PMID: 21320693). These papers should be discussed.

My quick alignment revealed that m139 harbours two short stretches (DTTYTLVREYITFR and DTAGELTPLGVCA) with certain sequence homology to the VACV-encoded proteins K7 and A52. Intriguingly, these similarities are either overlapping or close to regions, which are involved in DDX3X binding described in the following article (PMID: 19913487). More specific mutations might help to sort things out.

-SEM is not a descriptive statistics and should not be used as such. The replication curves and luciferase data sets should be depicted as mean (or median) ± standard deviation (SD).

-In line 229, there is a problem with the reference in the database.

-In Fig.2A, there seems to be an additional symbol (a black triangle?) at the 7 dpi point.

-All error bars in the replication curved should be depicted in a colour different from the curve and the data points in order to allow the reader to appreciate it.

Reviewer #3: In line 229 a reference is missing.

In line 24 the word “cell” is missing.

In the title of Fig. 1 the word “the” is duplicated.

PLOS authors have the option to publish the peer review history of their article (what does this mean?). If published, this will include your full peer review and any attached files.

Reviewer #1: No

Reviewer #2: No

Reviewer #3: No
---

## [Editor Report · Decision Letter 1]

9 Sep 2020

Dear Professor Brune,

We are pleased to inform you that your manuscript 'Murine cytomegaloviruses m139 targets DDX3 to curtail interferon production and promote viral replication' has been provisionally accepted for publication in PLOS Pathogens.

Best regards,

Ian R Humphreys

Guest Editor

PLOS Pathogens

Klaus Früh

Section Editor

PLOS Pathogens

Kasturi Haldar

Editor-in-Chief

PLOS Pathogens

orcid.org/0000-0001-5065-158X

Michael Malim

Editor-in-Chief

PLOS Pathogens

orcid.org/0000-0002-7699-2064

The authors have done an excellent job of addressing the core concerns, and have also addressed all of the addditonal concerns (with the exception of one comment involving complexities beyond the scope of this study).
---

## [Editor Report · Acceptance letter]

29 Sep 2020

Dear Dr. Brune,

We are delighted to inform you that your manuscript, "Murine cytomegaloviruses m139 targets DDX3 to curtail interferon production and promote viral replication," has been formally accepted for publication in PLOS Pathogens.

Best regards,

Kasturi Haldar

Editor-in-Chief

PLOS Pathogens

orcid.org/0000-0001-5065-158X

Michael Malim

Editor-in-Chief

PLOS Pathogens

orcid.org/0000-0002-7699-2064